# GSEA–SDBE: A gene selection method for breast cancer classification based on GSEA and analyzing differences in performance metrics

**Hu Ai** *

Department of Criminal Technology, Guizhou Police College, Guiyang, Guizhou, China

* HuAi10657@outlook.com

## Abstract

### Motivation

Selecting the most relevant genes for sample classification is a common process in gene expression studies. Moreover, determining the smallest set of relevant genes that can achieve the required classification performance is particularly important in diagnosing cancer and improving treatment.

### Results

In this study, I propose a novel method to eliminate irrelevant and redundant genes, and thus determine the smallest set of relevant genes for breast cancer diagnosis. The method is based on random forest models, gene set enrichment analysis (GSEA), and my developed Sort Difference Backward Elimination (SDBE) algorithm; hence, the method is named GSEA–SDBE. Using this method, genes are filtered according to their importance following random forest training and GSEA is used to select genes by core enrichment of Kyoto Encyclopedia of Genes and Genomes pathways that are strongly related to breast cancer. Subsequently, the SDBE algorithm is applied to eliminate redundant genes and identify the most relevant genes for breast cancer diagnosis. In the SDBE algorithm, the differences in the Matthews correlation coefficients (MCCs) of performing random forest models are computed before and after the deletion of each gene to indicate the degree of redundancy of the corresponding deleted gene on the remaining genes during backward elimination. Next, the obtained MCC difference list is divided into two parts from a set position and each part is respectively sorted. By continuously iterating and changing the set position, the most relevant genes are stably assembled on the left side of the gene list, facilitating their identification, and the redundant genes are gathered on the right side of the gene list for easy elimination. A cross-comparison of the SDBE algorithm was performed by respectively computing differences between MCCs and ROC_AUC_score and then respectively using 10-fold classification models, e.g., random forest (RF), support vector machine (SVM), k-nearest neighbor (KNN), extreme gradient boosting (XGBoost), and extremely randomized trees

**Citation:** Ai H (2022) GSEA–SDBE: A gene selection method for breast cancer classification based on GSEA and analyzing differences in performance metrics. PLoS ONE 17(4): e0263171. https://doi.org/10.1371/journal.pone.0263171

**Data Availability Statement:** Transcriptome datasets for breast, lung, and liver cancers and the clinical dataset of corresponding patients with breast cancer are available in TCGA database at https://portal.gdc.cancer.gov/repository. Its query

parameters are as follows: cases.primary_site in ["breast"] and cases.project.program.name in ["TCGA"] and files.data_category in ["transcriptome profiling"] and files.data_type in ["Gene Expression Quantification"]; cases.primary_site in ["bronchus and lung"] and cases.project.program.name in ["TCGA"] and files.data_category in ["transcriptome profiling"] and files.data_type in ["Gene Expression Quantification"]; cases.primary_site in ["liver and intrahepatic bile ducts"] and cases.project.program. name in ["TCGA"] and files.data_category in ["transcriptome profiling"] and files.data_type in ["Gene Expression Quantification"]. Genes expressed dataset for prostate cancer [44] can be found in the Broad Institute at https://www. broadinstitute.org/publications/broad12196. Gene expression dataset for colon cancer [45] are available in the Princeton University Gene Expression Project at http://genomics-pubs. princeton.edu/oncology/. The data that supports the findings of this study are available in the supplementary material of this article.

**Funding:** YES, Guizhou Province Science and Technology Planning Project (Qianke He [2016] Support 2847).

**Competing interests:** The authors have declared that no competing interests exist.

(ExtraTrees). Finally, the classification performance of the proposed method was compared with that of three advanced algorithms for five cancer datasets.

Results showed that analyzing MCC differences and using random forest models was the optimal solution for the SDBE algorithm. Accordingly, three consistently relevant genes (i.e., *VEGFD*, *TSLP*, and *PKMYT1*) were selected for the diagnosis of breast cancer. The performance metrics (MCC and ROC_AUC_score, respectively) of the random forest models based on 10-fold verification reached 95.28% and 98.75%. In addition, survival analysis showed that *VEGFD* and *TSLP* could be used to predict the prognosis of patients with breast cancer.

Moreover, the proposed method significantly outperformed the other methods tested as it allowed selecting a smaller number of genes while maintaining the required classification accuracy.

## Introduction

Selecting relevant genes to distinguish patients with or without cancer is a common task in gene expression research [1,2]. For genetic diagnosis in clinical practice, it is important to efficiently identify relevant genes and eliminate irrelevant and redundant genes to obtain the smallest possible gene set that can achieve good predictive performance [3].

To this end, genetic selection methods are of great importance. These methods can be roughly divided into three categories: filters, wrappers, and mixers [4]. In a previous study, I focused on a hybrid approach that combines the advantages of filter and wrapper methods [5]. For cancer classification, previous hybrid approaches have utilized symmetrical uncertainty to analyze the relevance of genes based on support vector machines [6], employed minimum redundancy and maximum relevance feature selection to select a subset of relevant genes [7], and applied Cuckoo search to select genes from microarray technology [8]. The hybrid approach essentially includes two processes, selecting relevant genes and eliminating redundant genes. To select relevant genes, previous research has utilized semantic similarity measurements of gene ontology terms based on definitions for similarity analysis of gene function [9], applied the concept of global and local gene relevance to calculate the equivalent principal component analysis load of nonlinear low-dimensional embedding [10], and obtained relevant features from the Cancer Genome Atlas (TCGA) transcriptome dataset by cooperative embedding [11]. Because relevant genes often contain redundant genes, the process of gene elimination is important for obtaining the minimal number of relevant genes that can function effectively in a classification model. Many methods can be applied including feature similarity estimated by explicitly building a linear classifier on each gene [12], homology searching against a gene or protein database [13], or the Cox-filter model [14].

In the present study, I propose a novel hybrid method that can determine the smallest set of relevant genes required to achieve accurate classification of breast cancer diagnosis. Breast cancer transcriptome data can be downloaded from the TCGA database; this unbalanced data was used in the current analyses. RF [15] and gene set enrichment analysis (GSEA) [16] were applied to select relevant breast cancer genes and the proposed Sort Difference Backward Elimination (SDBE) algorithm was then used to eliminate redundant genes from these relevant genes; hence, the proposed method was named GSEA–SDBE. First, a random forest model was constructed and trained with all the differential gene expression data and then the genes for which importance was almost zero were deleted. Subsequently, GSEA was applied to

analyze the remaining differentially expressed genes (DEGs) according to Kyoto Encyclopedia of Genes and Genomes (KEGG) pathway enrichment and those genes that were strongly related to breast cancer were selected from the enriched KEGG pathways. Then, the SDBE algorithm was applied to identify the important relevant genes from the selected genes. The SDBE algorithm includes a process by which the difference in the Matthews correlation coefficients (MCCs) of random forest models is calculated before and after the deletion of a given gene, which indicates the degree of redundancy of the corresponding deleted gene on the remaining genes according to backward elimination. Using the SDBE algorithm, the most relevant genes are stably collected on the left side of the gene list while the redundant genes are gathered on the right side of the gene list. Through the GSEA–SDBE method, an optimal model was created that could determine the smallest set of relevant genes for breast cancer diagnosis. Results showed that this method could achieve excellent classification performance for breast cancer. Furthermore, some of the selected relevant genes could be used to predict prognosis in patients with breast cancer.

## Materials and methods

### Data preparation

**Breast cancer transcriptome data.** Transcriptome data from breast cancer samples and the clinical data of corresponding patients were downloaded from TCGA database (https://gdc.cancer.gov/). A total of 1222 transcriptome samples, wherein each sample contained expression of 18584 genes, were obtained. This unbalanced dataset, which includes 113 normal and 1109 tumor tissues, was named BCT_1222 (113: 1109). In addition, the clinical data of 1109 patients with breast cancer were obtained.

**Differential expression analysis and normalization.** By performing the Mann–Whitney–Wilcoxon test in R software 3.6.2 (wilcox.tes) with $|logFC| > 1.0$ and p.FDR $< 0.05$ as the thresholds, 4579 DEGs were screened between the normal samples and tumor samples from the BCT_1222 dataset. These samples were randomly shuffled and the expression values of each DEG in all samples were respectively standardized via min–max normalization.

### Selecting genes by importance based on a random forest model

The random forest method can provide an assessment of variable importance to variable selection [17,18]. A random forest model was constructed and trained using Sklearn 0.22.2.post1 in python 3.6 with 4579 DEGs. The model was used to calculate the importance of variables (genes) and the genes were sorted by their importance in descending order. From these genes, a certain number of top genes were selected based on experience to reduce the burden of subsequent procedures.

### Gene selection by GSEA

GSEA [19] can be used to determine whether a group of genes shows statistically significant and concordant differences between two biological states according to enrichment analysis; here, it was performed by the JAVA program. The KEGG database includes a collection of manually drawn graphical maps known as KEGG pathway maps [20]. KEGG in the Molecular Signatures Database (MSigDB) [21] was chosen as the back-end database of GSEA. GSEA was run and genes were selected through the core enrichment [22] of KEGG pathways strongly related to breast cancer. Therefore, it was possible to screen for DEGs that were closely associated with breast cancer. Genes that were weakly associated with or were unrelated to breast cancer were filtered out, even if they had high importance in a random forest model.

## Metrics and benchmark methods

The performances of all classification models applied in this study were evaluated by 10-fold cross-validation. The models were trained and tested with 10-fold cross-validation. According to the prediction results and tested data, they were respectively merged in a given order. By comparing the prediction results with the tested data, true positives (TP), false positives (FP), false negatives (FN), and true negatives (TN) were obtained. Normal samples were negatives and tumor samples were positives. Tests were conducted on a real dataset with unbalanced data. Therefore, the effectiveness of the binary classification model was measured by several performance metrics [23] including accuracy (Acc), recall (Re), F1_score (F1), false positive rate (FPR), computed area under the receiver operating characteristic curve from prediction scores (ROC_AUC_score), and MCC. The formulas and functions are as follows:

$$ROC\_AUC\_score = sklearn.metrics.roc\_auc\_score \tag{1}$$

$$Acc = \frac{TN + TP}{TN + TP + FP + FN} \tag{2}$$

$$Re = \frac{TP}{TP + FN} \tag{3}$$

$$F1 = \frac{2 \times (Pr \times Re)}{Pr + Re} \tag{4}$$

$$FPR = \frac{FP}{FP + TN} \tag{5}$$

$$MCC = \frac{TP \times TN - FP \times FN}{\sqrt{(TP + FP) \times (TP + FN) \times (TN + FP) \times (TN + FN)}} \tag{6}$$

In addition, MCC [24,25] and ROC_AUC_score [26,27] are shown to better handle numerically unbalanced data sets.

## SDBE algorithm

The training, testing, and calculation of various performance metrics for all classification models were based on 10-fold cross-validation. The focus was on finding a high-performance classification model with the fewest variables (genes); subsequently, a novel algorithm, namely SDBE, was proposed. The underlying principle of the SDBE algorithm is that the performance metrics of the classification model will not change significantly after a redundant gene is deleted. Therefore, the differences in the chosen performance metrics were computed before and after deletion of each gene to indicate the degree of redundancy of the corresponding deleted gene on the remaining genes in backward elimination based on the random forest method. These deleted genes were collected into a list in reverse order during backward elimination [28].

From a set position, genes were sorted by their corresponding performance metric differences in descending order into the two parts and the two parts were then merged. Through continuously iterating and changing the set position, the important relevant genes were stably assembled on the left side of the gene list to facilitate their easy identification, whereas redundant genes were gathered on the right side of the gene list for easy elimination. The procedure

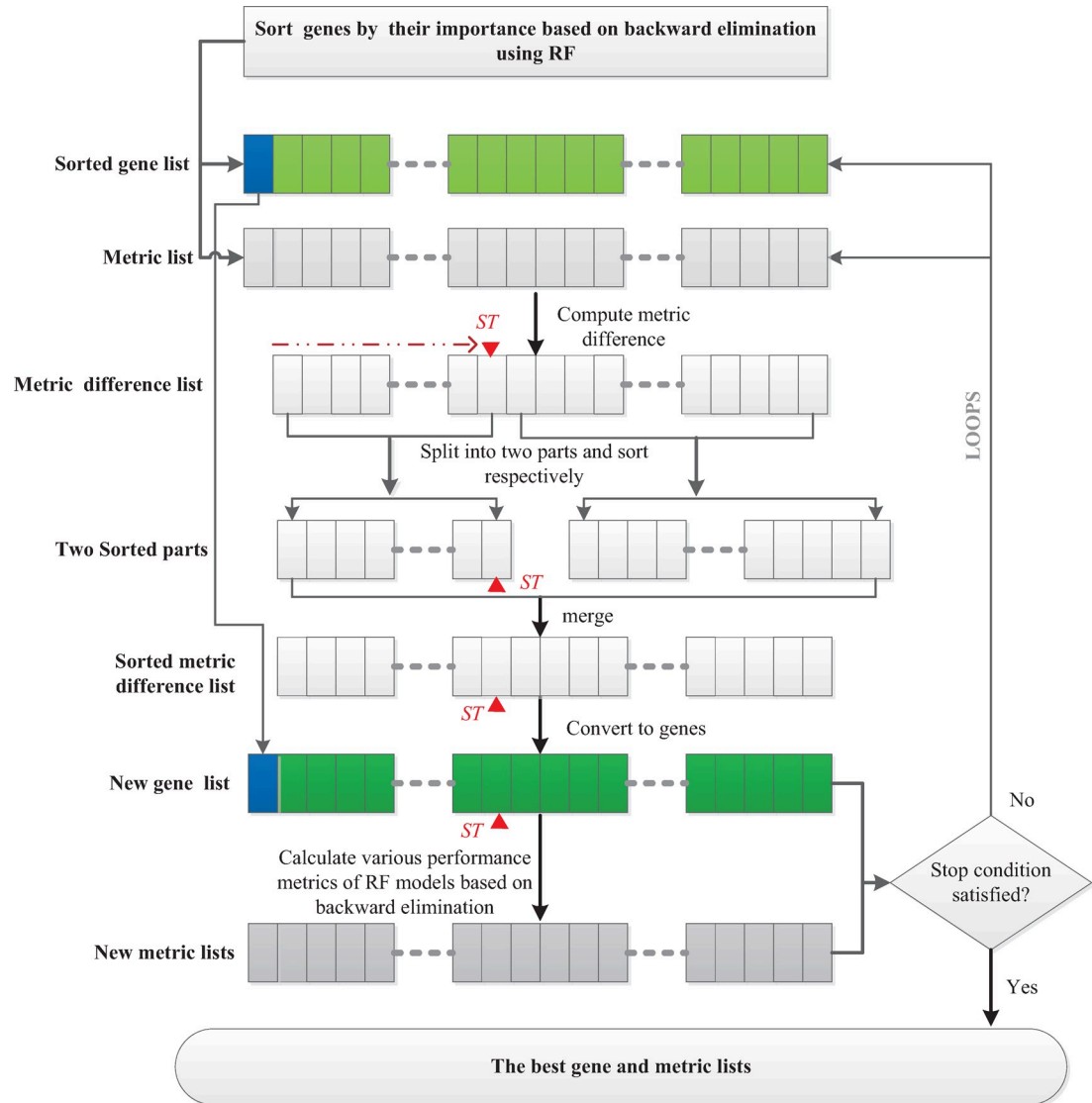

**Fig 1. Procedure of the Sort Difference Backward Elimination (SDBE) algorithm.**

underlying the SDBE algorithm is provided in Fig 1. The SDBE algorithm consists of seven stages as follows.

**Stage 1:** In each loop of backward elimination, 10-fold random forest models were trained and tested to calculate various performance metrics and the average importance of each variable, i.e., each gene. Next, these genes were sorted in descending order of average importance. After each loop of backward elimination, the deleted gene with the least importance and various metrics of the model were added to various dedicated lists. Thus, by respectively transposing all the lists, a list of genes $G(g_k, 0 \leq k \leq n)$ in descending order of importance and various metric lists were obtained. These lists were provided to the stages that followed. Importantly, gene $g_0$ at the first position in the list of the genes was determined at this stage because the position of this gene would not change in subsequent stages.

**Stage 2:** One of model performance metrics, such as MCC or ROC_AUC_score, was chosen as the object of difference analysis for subsequent stages and the index variable $ST$ was initialized to 0.

**Stage 3:** The following formula was used to compute the difference in the performance metric before and after gene deletion during backward elimination based on random forest modeling:

$$dm_i = m_i - m_{i-1}, 0 < i \le n, \tag{7}$$

where $m_i$ and $m_{i-1}$ respectively denote the metric before and after deleting gene $g_i (0 < i \le n)$ from sublist $Gs(g_u, 0 \le u \le i, 0 < i \le n)$ of gene list $G(g_k, 0 \le k \le n)$ in backward elimination. Only one gene was deleted from the end of list $Gs$ at each loop in backward elimination. The performance metric difference $dm_i (0 < i \le n)$ could indicate the degree of redundancy of the corresponding deleted gene $g_i (0 < i \le n)$ on the remaining genes of sublist $Gs$.

**Stage 4:** The value of the variable $ST$ was used as the index position to search forward in the metric difference list $DM(dm_i, 0 < i \le n)$ until an element $<0$ was encountered; the index of this element was used to update the variable $ST$.

**Stage 5:** The metric difference list $DM$ was split into two parts, part1 and part2 (including the element at index $ST$) by index $ST$, and then the elements in part1 and part2 were respectively sorted in descending order.

**Stage 6:** The elements of part1 and part2 were replaced with genes by the corresponding relationship between $dm_i (0 < i \le n)$ and $g_i (0 < i \le n)$, and then the two parts were merged into a new gene list $NG$. Subsequently, $g_0$ in the list $G$ was added to the end of the new list $NG$. Then, the list $NG$ was transposed.

**Stage 7:** The genes of the list $NG$ were analyzed by backward elimination. At each step of backward elimination, the 10-fold classification mode, e.g., random forest (RF), support vector machine (SVM), k-nearest neighbor (KNN), extreme gradient boosting (XGBoost), and extremely randomized trees (ExtraTrees), and ExtraTrees, was trained and tested to calculate various performance metrics. After each step of backward elimination, the performance metrics were respectively added to the corresponding metric lists. Then, the iteration was terminated and the data were saved. However, if the number of iterations set based on experience was not reached, the metrics lists, which were respectively transposed, and the list NG were sent to stage 3 to start a new iteration.

**Stage 8:** Mapping analysis of the metrics lists and the list NG was performed and the smallest set of relevant genes needed to achieve the required sample classification performance was determined.

## The entire pipeline of the GSEA–SDBE method

The gene selection procedure followed in the GSEA–SDBE method is provided in Fig 2.

## Results

### Differential expression analysis and normalization

From 4579 DEGs identified in the BT_1222 dataset, 2702 were upregulated and 1877 were downregulated. These genes are represented in a volcano plot in Fig 3.

### Random forest models

Having trained a random forest model with data on 4479 DEGs, the out-of-bag error was 0.01%. Genes were sorted by their importance in descending order, as shown in Fig 4. Selecting the top 2000 genes from the 4579 DEGs was optimal in the experiments; thus, the remaining 2579 genes, for which the importance was close to zero, were deleted.

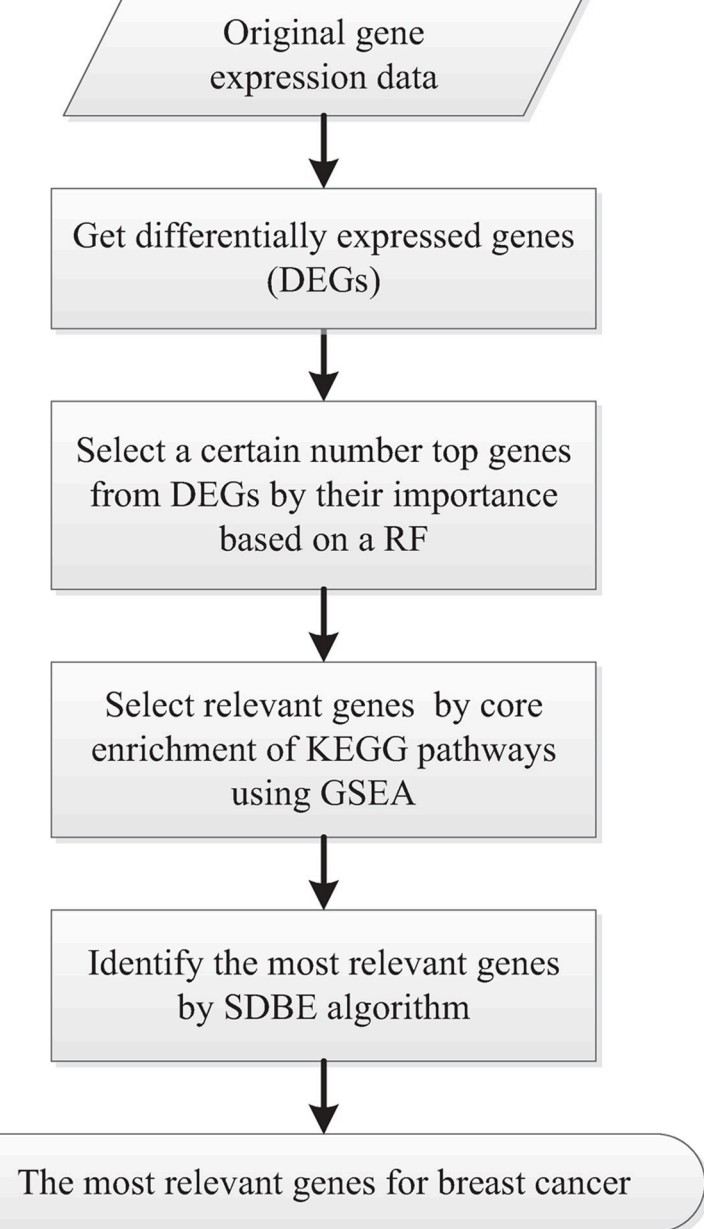

**Fig 2. Gene selection procedure in the GSEA–SDBE method.**

## GSEA

GSEA 3.0 was applied to analyze 2000 DEGs with KEGG pathways enrichment; the gene sets database was set to c2.cp.kegg.v7.1.symbols.gmt of the MSigDB. In enrichment results, 30 gene sets were obtained. These included five and 15 upregulated and downregulated gene sets in the phenotype "Tumor" (S1 Table), respectively. Four gene sets (Table 1) were selected that were strongly associated with breast cancer (Fig 5). Altogether, 60 genes were identified, including 20 upregulated genes and 40 downregulated genes, after deleting 12 repeated downregulated genes from 72 genes in the core enrichment of the four gene sets.

## Differential metabolites

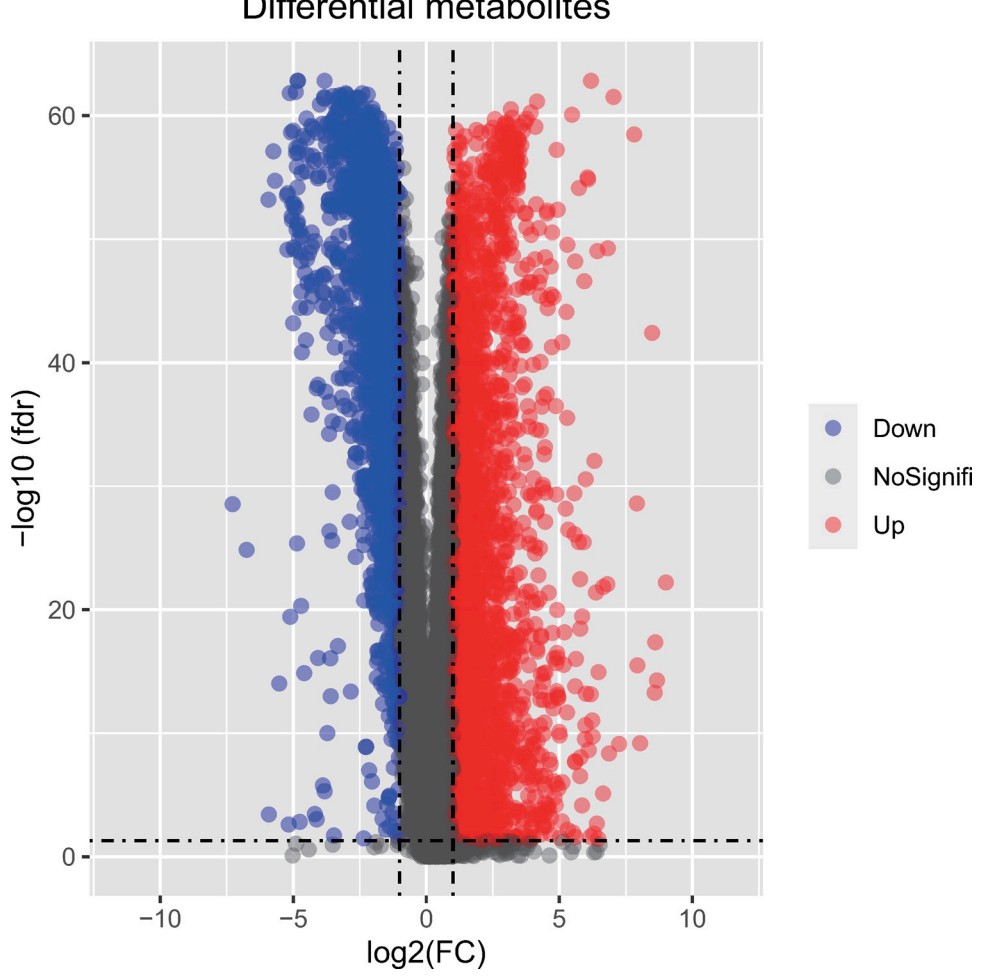

**Fig 3. Volcano plot of differentially expressed genes.** The red and blue dots represent upregulated and downregulated genes, respectively.

## SDBE algorithm

In the SDBE algorithm, the training, testing, and calculation of various performance metrics for all classification models were based on 10-fold cross-validation. The expression data of 60 genes from the GSEA enrichment analysis results were used in the SDBE algorithm. From stage 1 of the algorithm, 60 genes were listed in descending order of importance, as shown in S2 Table, and various metric lists (including Acc, Re, FPR, F1_score, ROC_AUC_score, and MCC) were illustrated using matplotlib in python 3.6 for comparison. It was difficult to select the smallest gene set that could still achieve good predictive performance by sorting genes by their importance, although ranking gene stages by importance was vital to the process. The most important part of this step was determining the top gene in the list as this gene does not change in subsequent stages. From this stage, the gene and metric lists were passed to the stages that followed.

In stage 2 of the SDBE algorithm, the performance metrics ROC_AUC_score and MCC were respectively chosen as the objects of difference analysis for subsequent iterations; each iteration included stage 3–7 and the number of iterations was set at 19. To compare the influence of different classification models in the SDBE algorithm, the following were respectively

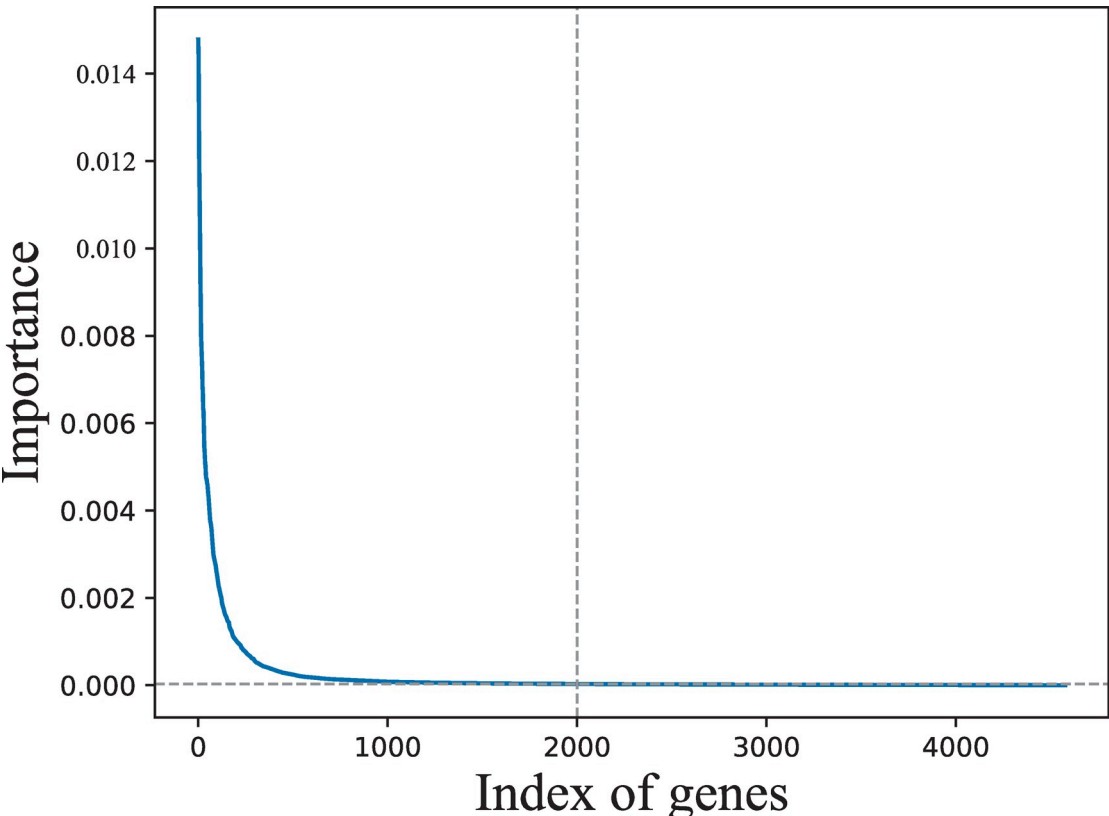

**Fig 4. Genes sorted by importance in descending order.**

chosen for use as the classification model: RF, SVM, KNN, XGBoost [29], and ExtraTrees [30]. Therefore, the SDBE algorithm was cross-tested. Regardless of the object chosen for difference analysis (ROC_AUC_score or MCC; Fig 6A and 6B) and the classification model (RF, SVM, KNN, XGBoost, or ExtraTrees) used, as the iteration progressed the most relevant genes were assembled in a stepwise manner on the left side of the gene list, whereas the redundant genes were gathered in a stepwise manner on the right side of the gene list (Fig 6). On the left side of the gene list, the identity and number of stable relevant genes differed depending on the analysis target and classification model, with three stable relevant genes being the maximum (S3 Table).

To cross-compare the SDBE algorithm, I used the 19th iterations of the algorithm and compared the same performance metrics of multiple classification models (RF, SVM, KNN, XGBoost, and ExtraTrees; Fig 6). As shown by the shapes of the polylines in Fig 7A, using

**Table 1. Gene sets (pathways) that were strongly related to breast cancer.**

| Gene set name | ES | NES | NOM P value | FDR Q value | Gene number (core enrichment) |
|---|---|---|---|---|---|
| KEGG_CELL_CYCLE | 0.60 | 1.37 | 0.201 | 0.319 | 20 |
| KEGG_CYTOKINE_CYTOKINE_RECEPTOR_INTERACTION | −0.29 | −0.96 | 0.496 | 0.726 | 17 |
| KEGG_JAK_STAT_SIGNALING_PATHWAY | −0.48 | −1.34 | 0.143 | 1.000 | 11 |
| KEGG_PATHWAYS_IN_CANCER | −0.23 | −0.84 | 0.720 | 0.790 | 24 |

ES: Enrichment score; NES: Normalized enrichment scores; NOM p-val: Nominal p value; FDR: False discovery rate.

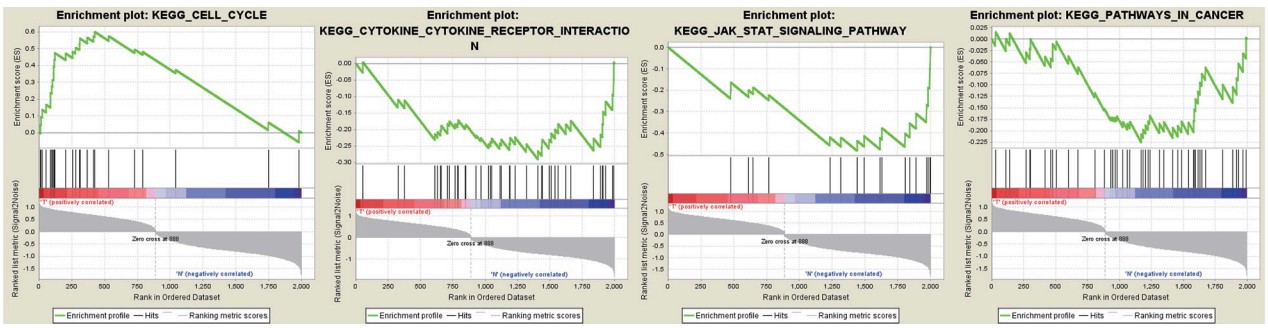

**Fig 5. Enrichment plots for the four gene sets (pathways) that were strongly related to breast cancer.**

MCC as the object of difference analysis produced better results than using ROC_AUC_score (Fig 7B). With MCC, the performance metrics of the RF model were better than the performance metrics of the other classification models; the blue polyline of the RF model was always

**Fig 6. Polylines of classification metrics, MCC, and ROC_AUC_score in 19 iterations.** (a) MCC as the object of difference analysis. (b) ROC_AUC_score as the object of difference analysis.

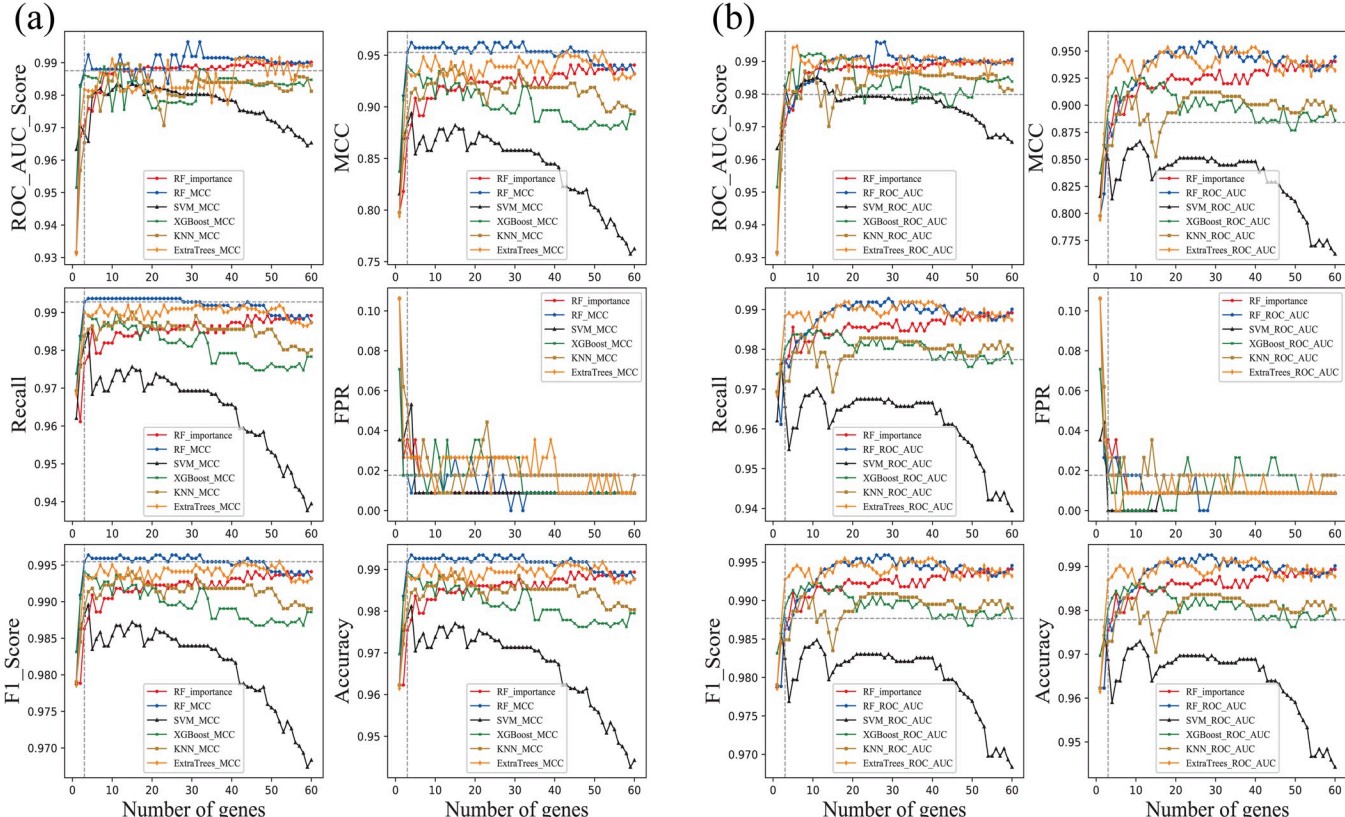

**Fig 7. Polylines of classification metrics at the 19th iteration of the Sort Difference Backward Elimination (SDBE) algorithm.** (a) MCC as the object of difference analysis. (b) ROC_AUC_score as the object of difference analysis. Various metric lists from stage 1 of the algorithm were illustrated by red polylines (RF_improtance).

above the other polylines. Therefore, I assessed the polyline of RF and found that the top three genes did not reach the peak or trough of the polyline but were close to each other (Fig 6A). More importantly, the top three genes were stable and repeatable. Therefore, I extracted performance metrics of classification models trained and tested using the top three genes from Fig 6 for comparison (Tables 2 and 3). Except for FPR (1.77%), the relative performance metrics of the RF model in Table 2, showing MCC as the object, were superior to those in Table 3 (ROC_AUC_score as the object); moreover, the top three genes from the classification models RF, KNN, XGBoost, and ExtraTrees were identical when MCC was the object (Table 2) but typically differed among the models when ROC_AUC_score was the object (Table 3). Because the data used to train and test the classification models were unbalanced (113 vs. 1109

**Table 2. MCC as the object of difference analysis: 10-fold cross-validation classification metrics of the top three genes.**

| Modes | ROC_AUC_score | MCC | Recall | FPR | F1_score | Accuracy | Top three genes |
|---|---|---|---|---|---|---|---|
| RF | 0.9875 | 0.9528 | 0.9928 | 0.0177 | 0.9955 | 0.9918 | VEGFD, TSLP, PKMYT1 |
| SVM | 0.9684 | 0.8832 | 0.9810 | 0.0442 | 0.9882 | 0.9787 | VEGFD, PKMYT1, BUB1B* |
| XGBoost | 0.9861 | 0.9396 | 0.9900 | 0.0177 | 0.9941 | 0.9893 | VEGFD, TSLP, PKMYT1 |
| KNN | 0.9653 | 0.8897 | 0.9837 | 0.0531 | 0.9891 | 0.9803 | VEGFD, TSLP, PKMYT1 |
| ExtraTrees | 0.9818 | 0.9345 | 0.9900 | 0.0265 | 0.9937 | 0.9885 | VEGFD, TSLP, PKMYT1 |

Genes marked with * are unstable genes in the SDBE algorithm.

**Table 3. ROC_AUC_score as the object of difference analysis: 10-fold cross-validation classification metrics of the top three genes.**

| Modes | ROC_AUC_score | MCC | Recall | FPR | F1_score | Accuracy | Top three genes |
|---|---|---|---|---|---|---|---|
| RF | 0.9799 | 0.8840 | 0.9774 | 0.0177 | 0.9877 | 0.9779 | VEGFD, SPRY2, BUB1B* |
| SVM | 0.9828 | 0.8501 | 0.9657 | 0.0 | 0.9825 | 0.9689 | VEGFD, CCNB1*, TSLP* |
| XGBoost | 0.9812 | 0.8952 | 0.9801 | 0.0177 | 0.9890 | 0.9803 | VEGFD, CCL14, TSLP |
| KNN | 0.9771 | 0.8627 | 0.9720 | 0.0177 | 0.9849 | 0.9710 | VEGFD, TSLP, CCL14 |
| ExtraTrees | 0.9809 | 0.9260 | 0.9883 | 0.0265 | 0.9927 | 0.9869 | VEGFD, TSLP, CDC25C |

Genes marked with * are unstable genes in the SDBE algorithm.

samples), the performance metrics MCC and ROC_AUC_score of the RF model were focused upon.

In summary, using MCC as the object of difference analysis and RF as the classification mode in the SDBE algorithm was optimal. In addition, three stable relevant genes, namely *VEGFD*, *TSLP*, and *PKMYT1*, were chosen for the diagnosis of breast cancer. Moreover, based on 10-fold verification, the performance metrics MCC and ROC_AUC_score for RF models were 95.28% and 98.75%, respectively.

## Survival analysis of patients

First, patients were divided into two groups, high and low risk, based on the median expression of a certain gene (S4 Table). If the gene was downregulated, the patients whose expression of the gene was lower than the median expression were classified as high risk, whereas the remaining patients were low risk. If the gene was upregulated, the method of grouping was reversed.

Kaplan–Meier survival analysis [31] and log-rank tests were used to determine the prognostic significance of expression of the three genes, *VEGFD*, *TSLP*, and *PKMYT1*, in patients with breast cancer. *VEGFD* and *TSLP* were downregulated genes, whereas *PKMYT1* was upregulated. A log-rank test revealed that patients with low *VEGFD* and *TSLP* expression had significantly shorter overall survival (OS) times than those patients with high expression of these genes (P = 0.0466 and P = 0.0003, respectively; Fig 8); the median OS times in months (with 95% confidence intervals) were 129 (114–142) and 116 (102–132), respectively; Fig 8 and Table 4). In contrast, the result of the log-rank test for *PKMYT1* was not significant

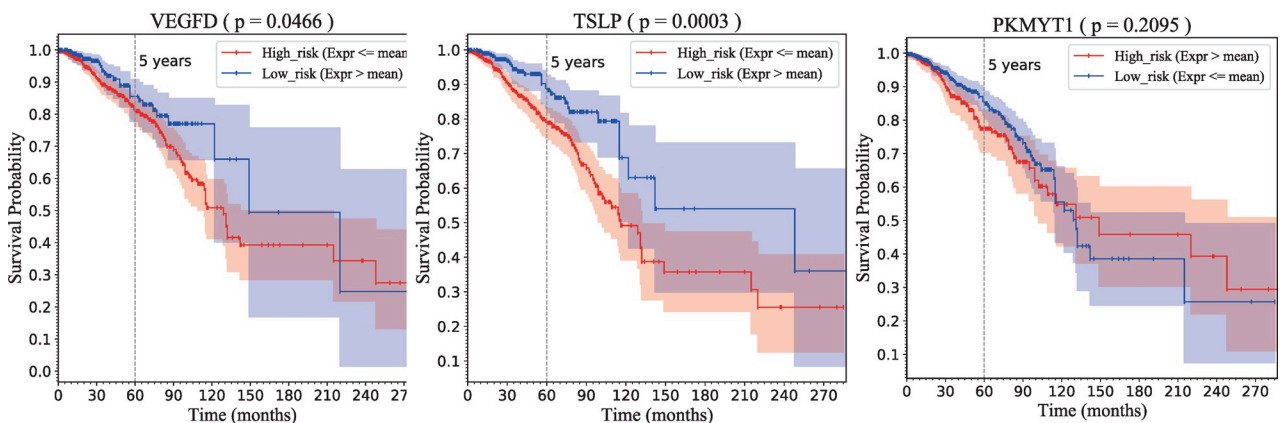

**Fig 8. Kaplan–Meier survival graphs for expression of VEGFD, TSLP, and PKMYT1.** Red and blue curves denote high-risk and low-risk groups, respectively.

Table 4. Results of survival analysis for high-risk and low-risk groups according to three genes.

| Gene name | Expression in tumor | P value | High risk | | | Low risk | | |
|---|---|---|---|---|---|---|---|---|
| | | | SP (5 y) | M-OS [95% CI] | N | SP (5 y) | M-OS [95% CI] | N |
| *VEGFD* | Downregulated | 0.0466 | 0.8088 | 129 [114–142] | 846 | 0.8552 | 149 [122–inf] | 262 |
| *TSLP* | Downregulated | 0.0003 | 0.7896 | 116 [102–132] | 786 | 0.8837 | 248 [122–inf] | 322 |
| *PKMYT1* | Upregulated | 0.2095 | 0.7743 | 149 [102–inf] | 419 | 0.8494 | 131 [115–215] | 689 |

P value: Comparison between high risk and low risk; Inf: Data points not obtained; SP (5 y): 5-year survival probability; M-OS (95% CI): Median overall survival time in months with 95% confidence intervals; N: Number of patients.

(P = 0.2095) and the polylines of the high-risk and low-risk groups for this gene crossed at 120 months (Fig 8). Therefore, *VEGFD* and *TSLP* could be used to predict prognosis in patients with breast cancer, whereas *PKMYT1* is not suitable for this purpose.

### Relevance of the selected genes to cancer

*VEGF-D* induces the formation of lymphatics within tumors, thereby facilitating the spread of the tumor to lymph nodes, and promotes tumor angiogenesis and growth [32–36]. *TSLP* is an interleukin-7 (IL-7)-like cytokine that is involved in the progression of various cancers and is a key mediator of breast cancer progression [37–40]. Human *PKMYT1* is an important regulator of the G2/M transition in the cell cycle. Studies have demonstrated that *PKMYT1* might be a therapeutic target in hepatocellular carcinoma and neuroblastoma [41–43].

### Performance comparison of GSEA–SDBE with that of other models

To test the feature selection performance of the GSEA–SDBE method, a simplified version, named Pre-SDBE, which does not use GSEA to filter out genes weakly associated with or unrelated to cancer, was used.

The three advanced gene selection algorithms were the genetic algorithm (GA), particle swarm optimization (PSO) algorithm, and cuckoo optimization algorithm and harmony search (COA-HS). These algorithms use 100 relevant genes selected via the minimum redundancy and maximum relevance (MRMR) as input data and the SVM as a classifier [7].

The classification performance of Pre-SDBE was compared with that of the three advanced algorithms for five cancer datasets composed of DEGs in breast, lung, and liver cancers and genes expressed in prostate and colon cancers (Table 5).

Table 5. Information on the datasets used for performance comparison.

| Name | Data sources | #Genes | #DEGs | #Samples | Normal | Tumor |
|---|---|---|---|---|---|---|
| **Breast** | TCGA [a] | 56,536 | 4,579 | 1,222 | 113 | 1,109 |
| **Lung** | TCGA [a] | 56,536 | 7,483 | 1,146 | 108 | 1,038 |
| **Liver** | TCGA [a] | 56,536 | 8,772 | 465 | 58 | 407 |
| **Prostate** | Microarray dataset [b] | 12,600 | – | 102 | 50 | 52 |
| **Colon** | Microarray dataset [c] | 7,457 | – | 62 | 22 | 40 |

[a] Database (https://gdc.cancer.gov/)

[b] Singh et al. [44]

[c] Alon et al. [45].

#Genes: Number of genes; #DEGs: Number of differentially expressed genes (obtained using wilcox.tes with |logFC| >1.0 and p.FDR <0.05); #Samples: Number of selected samples.

**Table 6. Classification metrics (%) of four optimization algorithms for five cancer datasets.**

| Algorithm | Breast | | | | | | Lung | | | | | |
|---|---|---|---|---|---|---|---|---|---|---|---|---|
| | #Genes | MCC | RA | F1 | SE | SP | #Genes | MCC | RA | F1 | SE | SP |
| **Pre-SDBE** | 4 | 98.07 | 99.42 | 99.82 | 99.73 | 99.12 | 3 | 97.45 | 98.93 | 99.76 | 99.71 | 98.15 |
| **PSO [a]** | 30 | 82.98 | 95.56 | 98.18 | 97.00 | 94.12 | 29 | 88.29 | 98.72 | 98.70 | 97.44 | 100 |
| **GA [a]** | 18 | 88.87 | 98.80 | 98.78 | 97.60 | 100 | 15 | 90.88 | 99.04 | 99.03 | 98.08 | 100 |
| **COA-HS [a]** | 11 | 90.93 | 97.78 | 99.09 | 98.50 | 97.06 | 8 | 89.56 | 98.88 | 98.87 | 97.76 | 100 |

| Liver | | | | | | Colon | | | | Prostate | | | |
|---|---|---|---|---|---|---|---|---|---|---|---|---|---|
| #Genes | MCC | RA | F1 | SE | SP | #Genes | AC | SE | SP | #Genes | AC | SE | SP |
| 3 | 96.98 | 98.12 | 99.63 | 99.75 | 96.49 | 2 | 100 | 100 | 100 | 5 | 98.99 | 98.99 | 98.99 |
| 24 | 62.03 | 91.87 | 91.15 | 83.74 | 100 | 11[a] | 96.42[a] | 85.80[a] | 100[a] | 19[a] | 98.04[a] | 91.80[a] | 100[a] |
| 16 | 68.30 | 93.90 | 93.51 | 87.80 | 100 | 14[a] | 95.16[a] | 84.60[a] | 100[a] | 28[a] | 98.04[a] | 91.80[a] | 100[a] |
| 9 | 72.73 | 95.12 | 94.87 | 90.24 | 100 | 5[a] | 100[a] | 100[a] | 100[a] | 5[a] | 100[a] | 100[a] | 100[a] |

[a] Elyasigomari et al. [7]; Pre-SDBE: Simplified version of the GSEA–SDBE method; RA: ROC_AUC_score; F1: F1_score; AC: Accuracy; SE: Sensitivity; SP: Specificity
#Genes: Number of selected genes.
Note: For unbalanced (breast, lung, and liver) and balanced data (colon and prostate), the performance metrics of the model are different.

In the step of the Pre-SDBE algorithm selecting genes by their importance, the top 50 relevant genes were selected based on a random forest model (S1 Fig). Next, these genes were fed into the SDBE algorithm to identify the most relevant genes with the highest accuracy. The number of iterations in the SDBE algorithm was set at 6, 7, 23, 3, and 10 for the breast, lung, liver, colon, and prostate cancer datasets, respectively. The Fitness of PSO, GA, and COA-HS over 100 iterations for each cancer dataset are shown in S2 Fig.

Table 6 shows that for unbalanced data (breast, lung, and liver cancers), the classification metrics (MCCs) of PSO, GA, and COA-HS algorithms were much lower than those of Pre-SDBE (98.07, 97.45, and 96.98 for breast, lung, and liver cancers, respectively). This indicated that the PSO, GA, and COA-HS algorithms did not perform well for unbalanced data.

For the five cancer datasets, whether the data were balanced or unbalanced, Pre-SDBE outperformed the other three algorithms, achieving the highest classification accuracy while identifying fewer number of genes (Table 6). More details are shown in S3 Fig, S5 and S6 Tables.

## Discussion

In this study, DEGs were extracted from a breast cancer data set. Genes that are not significantly differentially expressed but have important biological significance for breast cancer could easily be missed in this process; however, even if these lost genes are retained, they may be deleted in subsequent processing. Indeed, such genes would be ignored by the classification model used in the GSEA–SDBE method described here. Nevertheless, this did not affect the ability of the method to identify some key genes for the diagnosis of breast cancer.

Dimensionality reduction runs through the entire GSEA–SDBE method; each step in the method prepares for dimensionality reduction in the next step. According to experience, selecting too few genes leads to some important pathways not being enriched, whereas selecting too many genes overfills the core enrichment of pathways with genes that make subsequent gene elimination difficult and GSEA time consuming. Therefore, the list of DEGs was sorted in descending order by variable importance according to a random forest model; the top 2000 genes were selected for analysis and some genes with importance close to zero were removed based on experience.

Although the selection of KEGG pathways in GSEA based on experience is subjective, it does not prevent obvious DEGs with no important biological significance for breast cancer being filtered out. In addition, these genes may also enhance the performance of classification models and the selection of important genes would be compromised. To eliminate redundant genes from the selected genes, the SDBE algorithm was applied. This algorithm computed the difference in performance metrics of the classification model before and after gene deletion during backward elimination, which indicated the degree of redundancy of the deleted gene on the remaining genes. When a gene was deleted from the gene list in this manner, the performance metrics of the classification model did not change significantly. Therefore, the deleted gene was similar to some remaining genes, and thus considered redundant.

Given the underlying principle of the SDBE algorithm, the top gene in the gene list would not participate in the sorting process and would not be recognized as redundant; additionally, the first gene in a similar gene group in the gene list would not be recognized as redundant or deleted. Therefore, stage 1 of the SDBE algorithm is particularly important because genes are sorted by their importance in RF during backward elimination at this stage.

At stage 5 of the SDBE algorithm, to speed up the sorting process and reduce the number of cycles, the metric difference list was divided into two parts from a set position and these two parts were respectively sorted in descending order. The change of the set position occurred at stage 4. From the set position in the metric difference list, a forward search was conducted until an element with a value less than the threshold, which was set at zero, was encountered; the index of this element was used to update the set position. If the threshold was set to a certain value greater than zero, this may be more conducive to sorting. However, from the 19 iterations shown Figs 2 and 3, the polylines of the performance metrics for the classification models, particularly RF with MCC as the object of difference analysis, met the requirements. Including many more iterations would have been more time consuming. However, setting ROC_AUC_score as the object of difference analysis was less effective compared with using MCC, which might be related to the complexity of the ROC_AUC_score formula.

In contrast to Pre-SDBE, the three advanced algorithms (GA, PSO, and COA-HS) did not filter out genes without biological significance for cancer and were much more time-consuming. This is likely because the three algorithms used MRMR to select input genes (S6 Table). Selecting fewer than 50 genes by their importance based on a random forest model as the input to the SDBE algorithm might save time. However, the 10-fold cross-validation was the main time-consuming factor in the GSEA–SDBE method and its simplified version (Pre-SDBE).

Here, the proposed GSEA–SDBE method was used to analyze breast cancer datasets. It allowed determining the smallest set of biologically relevant genes for cancer diagnosis. The simplified GSEA–SDBE method (Pre-SDBE) was used to select genes to classify cancer datasets to test the feature selection performance of GSEA–SDBE. The results showed that the GSEA–SDBE and Pre-SDBE methods were excellent. In the future, I will apply the GSEA–SDBE method to many types of cancer data and Pre-SDBE to feature selection for various types of data.

## Supporting information

**S1 Fig. Genes sorted by importance in descending order (Pre-SDBE).**
(TIF)

**S2 Fig. Fitness over 100 iterations for breast, lung, and liver cancers (PSO, GA, and COA-HS).**
(TIF)

**S3 Fig. Polylines of classification metrics of the Sort Difference Backward Elimination (SDBE) algorithm (Pre-SDBE).**
(TIF)

**S1 Table. Gsea_report_for_Tumor_and_Normal.**
(XLS)

**S2 Table. The 60 genes listed in descending order of importance.**
(XLSX)

**S3 Table. Genes sorted in a descending order in 19 iterations.**
(XLS)

**S4 Table. Information about survival of patients.**
(XLS)

**S5 Table. Genes sorted by SDBE algorithm in descending order (Pre_SDBE).**
(XLSX)

**S6 Table. Classification performance information of three advanced algorithms (PSO, GA, and COA-HS) for three cancer datasets.**
(DOCX)

**S1 Graphical abstract.**
(TIF)

## Acknowledgments

The author thanks the TCGA database for providing free data and allowing free usage of GSEA.

## Author Contributions

**Writing – original draft:** Hu Ai.

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
