## [Decision Letter · Decision Letter 0]

17 Jun 2021

PONE-D-21-09636

GSEA–SDBE: A gene selection method for breast cancer classification based on GSEA and analyzing differences in performance metrics

PLOS ONE

Dear Dr. Ai,

Thank you for submitting your manuscript to PLOS ONE. After careful consideration, we feel that it has merit but does not fully meet PLOS ONE’s publication criteria as it currently stands. Therefore, we invite you to submit a revised version of the manuscript that addresses the points raised during the review process.

We look forward to receiving your revised manuscript.

Kind regards,

Khanh N.Q. Le

Academic Editor

PLOS ONE

Journal Requirements:

Reviewers' comments:

Reviewer's Responses to Questions

**Comments to the Author**

1. Is the manuscript technically sound, and do the data support the conclusions?

Reviewer #1: Yes

Reviewer #2: Partly

2. Has the statistical analysis been performed appropriately and rigorously? 

Reviewer #1: Yes

Reviewer #2: No

3. Have the authors made all data underlying the findings in their manuscript fully available?

Reviewer #1: Yes

Reviewer #2: Yes

4. Is the manuscript presented in an intelligible fashion and written in standard English?

Reviewer #1: Yes

Reviewer #2: Yes

5. Review Comments to the Author

Reviewer #1: The manuscript is well written and the subject is very interesting. It has been written clearly and understandable.

I think if the author provides a practical flowchart for readers it would be more applicable. Also please provide some information about survival of the patients in the results section.

Reviewer #2: The paper does contribute to body of the existing knowledge. It needs some more work for further improvement.

My main concerns are:

1. The paper is missing a comparison with other state-of-the-art methods for gene selection.

2. A discussion on the computational complexity of the method after comparison with the other methods is missing.

Some minor comments.

i) The paper needs language corrections.

ii) References are not coherent. Some references are missing authors' names (et al. should not be in the bibliography)

iii) Add mathematical description of the methods covered. Algorithm are messy, please make them clear and easy to understand.

6. PLOS authors have the option to publish the peer review history of their article (what does this mean?). If published, this will include your full peer review and any attached files.

Reviewer #1: No

Reviewer #2: No

---

## [Author Response · Author response to Decision Letter 0]

28 Aug 2021

Responds to the reviewer’s commants:

Reviewer #1:

1. Response to comment: ( I think if the author provides a practical flowchart for readers it would be more applicable.)

Response: I am very sorry for my negligence. The revised practical flowcharts are as follows.

Fig. 1. Procedure of the Sort Difference Backward Elimination (SDBE) algorithm.

Fig 2. Gene selection procedure in the GSEA–SDBE method.

2. Response to comment : (Also please provide some information about survival of the patients in the results section.)

Response: I am very sorry for my negligence of providing some information about survival of the patients. So I added the S4 table, which provided this information.

Reviewer #2:

1. Response to comment :( The paper is missing a comparison with other state-of-the-art methods for gene selection.)

Response: I am very sorry for my negligence. The performance comparison with other models as follows.

Performance comparison with other models

The Pre-SDBE is a simplified version of the GSEA-SDBE method, which does not contain the use of GSEA to filter out genes weakly associated with or unrelated to cancer. The other three state-of-the-art-methods [7], the genetic algorithm (GA), the particle swarm optimization (PSO) algorithm, and the cuckoo optimization algorithm and harmony search (COA-HS), also does not filter out DEGs that have no biological significance for cancer. Therefore, the performance of the Pre-SDBE can be compared with these three algorithms which use relevant genes selected by the minimum redundancy and maximum relevance (MRMR) as input data.

In the Pre-SDBE algorithm, the top 50 relevant genes from DEGs were selected in the step that selected genes by their importance based on a random forest model (S1 Fig). Next, these selected genes were fed into SDBE algorithm to pick the best genes while maintaining the highest accuracy. The number of iterations in SDBE algorithm was respectively set at 3 and 10 for the colon and prostate cancer dataset (S5 Table).

Table 5. Information on the microarray datasets used in this comparison

Microarray dataset Number of genes Number of samples Normal Tumor

Prostate [32] 12,600 102 50 52

Colon [33] 7,457 62 22 40

In the case of the colon cancer dataset, the Pre-SDBE method outperformed other algorithms, reaching 100% accuracy with the fewest number of genes (2 genes) compared to all other algorithms (S2 Fig,). For the prostate cancer dataset, the Pre-SDBE method could also achieve accuracy (98.99%) with the fewest genes (5 genes) (Table 6 and S3 Fig).

Table 6. Performance comparison of Pre-SDBE with three models for two microarray datasets

Algorithm Colon Prostate

 #Genes AC SE SP #Genes AC SE SP

Pre-SDBE 2 1.0000 1.0000 1.0000 5 0.9899 0.9899 0.9899

PSO [7] 11 0.9642 0.8580 1.0000 19 0.9804 0.9180 1.0000

GA1 [7] 14 0.9516 0.8460 1.0000 28 0.9804 0.9180 1.0000

COA-HS [7] 5 1.0000 1.0000 1.0000 5 1.0000 1.0000 1.0000

AC:Accuracy; SE:Sensitivity; SP:Specificity; Number of selected genes: #Genes.

2. Response to comment : (A discussion on the computational complexity of the method after comparison with the other methods is missing.)

Response: I am very sorry for my negligence. The discussion about computational complexity as follows.

In the Pre-SDBE method, the top 50 relevant genes were first selected by their importance based on a random forest model, and its number was only half of the number of genes selected using MRMR [7]. Perhaps fewer genes could be selected as the input to the SDBE algorithm to reduce time-consuming. However, whether it is GSEA-SDBE method or its simplified version (Pre-SDBE), 10-fold cross-validation is the main factor leading to time-consuming.

Some minor comments

1. The paper needs language corrections. 

Response: I am very sorry for my incorrect writing. So, I have seriously corrected the language. Please check the revised paper.

2. References are not coherent. Some references are missing authors' names (et al. should not be in the bibliography)

Response: I am very sorry for my incorrect writing and revised the references as follows.

References

1. Hartmaier R, Albacker LA, Chmielecki J, Bailey M, He J, Goldberg ME, et al.. High-throughput genomic profiling of adult solid tumors reveals novel insights into cancer pathogenesis. Cancer Research. 2017;77:2464–2475. doi: 10.1158/0008-5472.CAN-16-2479.

2. Giovannantonio MD, Harris BH, Zhang P, Kitchen­Smith I, Xiong L, Sahgal N, et al. Heritable genetic variants in key cancer genes link cancer risk with anthropometric traits. Journal of Medical Genetics. 2020;0:1–8. doi: 10.1136/jmedgenet-2019-106799.

3. Dı´az-Uriarte R, Andre´s SAd. Gene selection and classification of microarray data using random forest. BMC Bioinformatics. 2006;7(3):1–13. doi: 10.1186/1471-2105-7-3.

4. Pok G, Liu J-CS, Ryu KH. Effective feature selection framework for cluster analysis of microarray data. Bioinformation. 2010; 4(8):385–389. doi: 10.6026/97320630004385.

5. Xie J, Wang C. Using support vector machines with a novel hybrid feature selection method for diagnosis of erythemato-squamous diseases. Expert Syst Appl. 2011; 38(5): 5809–5815. doi:10.1016/j.eswa.2010.10.050.

6. Piao Y, Piao M, Park K, Ryu KH. An ensemble correlation-based gene selection algorithm for cancer classification with gene expression data. Bioinformatics. 2012; 28(24): 3306–3315. doi:10.1093/bioinformatics/bts602.

7. Elyasigomari V, Lee DA, Screen HRC, Shaheed MH. Development of a two-stage gene selection method that incorporates a novel hybrid approach using the cuckoo optimization algorithm and harmony search for cancer classification. Journal of Biomedical Informatics. 2017; 67:11–20. doi:10.1016/j.jbi.2017.01.016.

8. Sampathkumar A, Rastogi R, Arukonda S, Shankar A, Kautish S, Sivaram M. An efficient hybrid methodology for detection of cancer-causing gene using CSC for micro array data. J Ambient Intell Humaniz Comput. 2020; 11(3):4743–4751. doi:10.1007/s12652-020-01731-7.

9. Pesaranghader A, Matwin S, Sokolova M, Beiko RG. SimDEF: definition-based semantic similarity measure of gene ontology terms for functional similarity analysis of genes. Bioinformatics 2016; 32(9): 1380–1387. doi: 10.1093/bioinformatics/btv755.

10. Angerer P, Fischer DS, Theis FJ, Scialdone A, Marr C. Automatic identification of relevant genes from low-dimensional embeddings of single-cell RNA-seq data. Bioinformatics. 2020;36(15):4291–4295. doi: 10.1093/bioinformatics/btaa198.

11. Kuang S, Wei Y, Wang L. Expression-based prediction of human essential genes and candidate lncRNAs in cancer cells. Bioinformatics. 2021; 37(3):396–403. doi:10.1093/bioinformatics/btaa717.

12. Zeng XQ, Li GZ, Yang JY, Yang MQ, Wu GF. Dimension reduction with redundant gene elimination for tumor classification. BMC Bioinformatics 2008; 9 (Suppl 6): S8. doi:10.1186/1471-2105-9-S6-S8.

13. Ono H, Ishii K, Kozaki T, Ogiwara I, Kanekatsu M, Yamada T. Removal of redundant contigs from de novo RNA-Seq assemblies via homology search improves accurate detection of differentially expressed genes. BMC Genomics. 2015; 16(1):1031–1044. doi: 10.1186/s12864-015-2247-0.

14. Suyan T. Identification of subtypespecific prognostic signatures using Cox models with redundant gene elimination. Oncology Letters. 2018; 15:8545-8555. doi: 10.3892/ol.2018.8418.

15. Pashaei E, Aydin N. Binary black hole algorithm for feature selection and classification on biological data. Applied Soft Computing. 2017;56,94–106. doi: 10.1016/j.asoc.2017.03.002.

16. Xiao Y, Hsiao T-H, Suresh U, Chen H-IH, Wu X, Wolf SE, et al. A novel significance score for gene selection and ranking. Bioinformatics. 2014;30(6):801–807. doi:10.1093/bioinformatics/btr671.

17. Deng H, Runger G.. Gene selection with guided regularized random forest. Pattern Recognition. 2013; 46(12): 3483–3489. doi: 10.1016/j.patcog.2013.05.018.

18. Alikovi E, Subasi A. Breast cancer diagnosis using GA feature selection and Rotation Forest. Neural Computing and Applications. 2017;28(4):753–763. doi: 10.1007/s00521-015-2103-9.

19. Subramanian A, Kuehn H, Gould J, Tamayo P, Mesirov JP. GSEA-P: a desktop application for gene set enrichment analysis. Bioinformatics. 2007; 23(23):3251–3253. doi:10.1093/bioinformatics/btm369.

20. Ogata H, Goto S, Sato K, Fujibuchi w, Bono H, Kanehisa M. KEGG: kyoto Encyclopedia of Genes and Genomes. Nucleic Acids Research. 1999;27(1):29–34. doi:10.1093/nar/27.1.29.

21.Liberzon A, Subramanian A, Pinchback R, Thorvaldsdóttir H, Tamayo p, Mesirov JP. Molecular signature database (msigdb) 3.0. Bioinformatics. 2011; 27(12):1739–1740. doi:10.1093/bioinformatics/btr260.

22. Reimand J, Isserlin R, Voisin V, Kucera M, Tannus-Lopes C, Rostamianfar A, et al. Pathway enrichment analysis and visualization of omics data using g:Profiler, GSEA, Cytoscape and EnrichmentMap. Nature Protocols. 2019; 14(2): 482–517. doi:10.1038/s41596-018-0103-9.

23. Robinson D. The statistical evaluation of medical tests for classification and prediction by m. sullivan pepe. Appl Stat. 2010;169(3): 656–656. doi: 10.1111/j.1467-985X.2006.00430_9.x

24. Khoury P, Gorse D. Investing in emerging markets using neural networks and particle swarm optimisation. International Joint Conference on Neural Networks. IEEE. 2015;1-7. doi:10.1109/IJCNN.2015.7280777.

25. Boughorbel S, Jarray F, El-Anbari M. Optimal classifier for imbalanced data using Matthews correlation coefficient metric. PLoS One. 2017;12(6):e0177678. doi: 10.1371/journal.pone.0177678.

26. Chawla NV, Karakoulas G. Learning from labeled and unlabeled data: an empirical study across techniques and domains. Journal of Artificial Intelligence Research. 2005; 23:331–366. doi:10.1613/jair.1509.

27. Fawcett T. An introduction to ROC analysis. Pattern Recognit Lett. 2006; 27(8): 861–874.

28. John GH, Kohavi R, Pfleger K. Irrelevant Features and the Subset Selection Problem. Machine Learning Proceedings 1994; 121–129. doi: 10.1016/B978-1-55860-335-6.50023-4.

29. Chen T, Guestrin C. XGBoost: a scalable tree boosting system. In: the proceedings of 22nd ACM SIGKDD conference on knowledge discovery and data mining. ACM.New York’ KDD. 2016; 785–794.

30. Geurts P, Ernst D. Wehenkel L. Extremely randomized trees. Machine Learning. 2006;63(1):3–42. doi: 10.1007/s10994-006-6226-1.

31. Foldvary N, Nashold B, Mascha E, Thompson EA, Lee N, McNamara JO, et al. Seizure outcome after temporal lobectomy for temporal lobe epilepsy: a kaplan-meier survival analysis. Neurology. 2000; 54(3):630–634. doi: 10.1212/WNL.54.3.630.

32. Singh D, Febbo PG, Ross K, Jackson DG, Manola J, Ladd C, et al. Gene expression correlates of clinical prostate cancer behavior, Cancer Cell. 2002;1(2):203–209. doi:10.1016/S1535-6108(02)00030-2.

33. Alon U, Barkai N, Notterman DA, Gish K, Ybarra S, Mack D, et al. Broad patterns of gene expression revealed by clustering analysis of tumor and normal colon tissues probed by oligonucleotide arrays. Proc. Natl. Acad. Sci. USA. 1999; 96 (12): 6745–6750. doi:10.1073/pnas.96.12.6745.

3. Add mathematical description of the methods covered. Algorithm are messy, please make them clear and easy to understand.

Response: I am very sorry. I'm afraid I don't have this ability yet, but I will work hard in the future.

---

## [Decision Letter · Decision Letter 1]

25 Oct 2021

PONE-D-21-09636R1GSEA–SDBE: A gene selection method for breast cancer classification based on GSEA and analyzing differences in performance metricsPLOS ONE

Dear Dr. Ai,

Thank you for submitting your manuscript to PLOS ONE. After careful consideration, we feel that it has merit but does not fully meet PLOS ONE’s publication criteria as it currently stands. Therefore, we invite you to submit a revised version of the manuscript that addresses the points raised during the review process.

We look forward to receiving your revised manuscript.

Kind regards,

Khanh N.Q. Le

Academic Editor

PLOS ONE

Journal Requirements:

Reviewers' comments:

Reviewer's Responses to Questions

**Comments to the Author**

1. If the authors have adequately addressed your comments raised in a previous round of review and you feel that this manuscript is now acceptable for publication, you may indicate that here to bypass the “Comments to the Author” section, enter your conflict of interest statement in the “Confidential to Editor” section, and submit your "Accept" recommendation.

Reviewer #2: (No Response)

2. Is the manuscript technically sound, and do the data support the conclusions?

Reviewer #2: Partly

3. Has the statistical analysis been performed appropriately and rigorously? 

Reviewer #2: Yes

4. Have the authors made all data underlying the findings in their manuscript fully available?

Reviewer #2: No

5. Is the manuscript presented in an intelligible fashion and written in standard English?

Reviewer #2: No

6. Review Comments to the Author

Reviewer #2: I suggest the following for further improving the article

1. The article is lacking an academic way of writing. It needs a thorough revision. Improve language and presentation of the paper.

2. Please consider various breast cancer datasets for showing the efficacy of the method.

3. Add redundancy assessment measures in addition to MCC.

4. Provide biological significance of the selected genes by various methods.

5. Provide experimental setup for the analysis in more details. What were the criteria used for parameter selection.

7. PLOS authors have the option to publish the peer review history of their article (what does this mean?). If published, this will include your full peer review and any attached files.

Reviewer #2: No

---

## [Author Response · Author response to Decision Letter 1]

25 Dec 2021

Responds to the reviewer’s commants:

Reviewer #2:

1. Response to comment: (The article is lacking an academic way of writing. It needs a thorough revision. Improve language and presentation of the paper).

Response: I am very sorry for my incorrect writing. So, I have seriously corrected the language. Please check the revised paper.

2. Response to comment: (Please consider various breast cancer datasets for showing the efficacy of the method)

Response: I am very sorry for my negligence. To show the efficacy of Pre-SDBE method, the classification performance of this method was compared with that of the three advanced algorithms for five cancer datasets (breast, lung, liver, colon, and prostate cancer datasets).

Performance comparison with other models

 The Pre-SDBE is a simplified version of the GSEA-SDBE method, which does not contain the use of GSEA to filter out genes weakly associated with or unrelated to cancer. The other three state-of-the-art-methods [7], the genetic algorithm (GA), the particle swarm optimization (PSO) algorithm, and the cuckoo optimization algorithm and harmony search (COA-HS), also does not filter out DEGs that have no biological significance for cancer. Therefore, the performance of the Pre-SDBE can be compared with these three algorithms which use relevant genes selected by the minimum redundancy and maximum relevance (MRMR) as input data.

 In the Pre-SDBE algorithm, the top 50 relevant genes from DEGs were selected in the step that selected genes by their importance based on a random forest model (S1 Fig). Next, these selected genes were fed into SDBE algorithm to pick the best genes while maintaining the highest accuracy. The number of iterations in SDBE algorithm was respectively set at 3 and 10 for the colon and prostate cancer dataset (S5 Table).

Table 5. Information on the microarray datasets used in this comparison

Microarray dataset Number of genes Number of samples Normal Tumor

Prostate [32] 12,600 102 50 52

Colon [33] 7,457 62 22 40

In the case of the colon cancer dataset, the Pre-SDBE method outperformed other algorithms, reaching 100% accuracy with the fewest number of genes (2 genes) compared to all other algorithms (S2 Fig,). For the prostate cancer dataset, the Pre-SDBE method could also achieve accuracy (98.99%) with the fewest genes (5 genes) (Table 6 and S3 Fig).

Table 6. Performance comparison of Pre-SDBE with three models for two microarray datasets

Algorithm Colon Prostate

 #Genes AC SE SP #Genes AC SE SP

Pre-SDBE 2 1.0000 1.0000 1.0000 5 0.9899 0.9899 0.9899

PSO [7] 11 0.9642 0.8580 1.0000 19 0.9804 0.9180 1.0000

GA1 [7] 14 0.9516 0.8460 1.0000 28 0.9804 0.9180 1.0000

COA-HS [7] 5 1.0000 1.0000 1.0000 5 1.0000 1.0000 1.0000

Pre-SDBE: A simplified version of the GSEA-SDBE method; AC: Accuracy; SE: Sensitivity; SP:Specificity; #Genes: Number of selected gene

Performance comparison of GSEA–SDBE with that of other models

To test the feature selection performance of the GSEA–SDBE method, a simplified version, named Pre-SDBE, which does not use GSEA to filter out genes weakly associated with or unrelated to cancer, was used.

The three advanced gene selection algorithms were the genetic algorithm (GA), particle swarm optimization (PSO) algorithm, and cuckoo optimization algorithm and harmony search (COA-HS). These algorithms use 100 relevant genes selected via the minimum redundancy and maximum relevance (MRMR) as input data and the SVM as a classifier [7].

The classification performance of Pre-SDBE was compared with that of the three advanced algorithms for five cancer datasets composed of DEGs in breast, lung, and liver cancers and genes expressed in prostate and colon cancers (Table 5).

Table 5. Information on the datasets used for performance comparison.

Name Data sources #Genes #DEGs #Samples Normal Tumor

Breast TCGA a 56,536 4,579 1,222 113 1,109

Lung TCGA a 56,536 7,483 1,146 108 1,038

Liver TCGA a 56,536 8,772 465 58 407

Prostate Microarray dataset b 12,600 － 102 50 52

Colon Microarray dataset c 7,457 － 62 22 40

a Database (https://cancergenome.nih.gov/); b Singh et al. [44]; c Alon et al. [45];

#Genes: number of genes; #DEGs: number of differentially expressed genes (obtained using wilcox.tes with logFC >1 and FDR <0.05); #Samples: number of selected samples.

In the step of the Pre-SDBE algorithm selecting genes by their importance, the top 50 relevant genes were selected based on a random forest model (S1 Fig). Next, these genes were fed into the SDBE algorithm to identify the most relevant genes with the highest accuracy. The number of iterations in the SDBE algorithm was set at 6, 7, 23, 3, and 10 for the breast, lung, liver, colon, and prostate cancer datasets, respectively. The Fitness of PSO, GA, and COA-HS over 100 iterations for each cancer dataset are shown in S2 Fig.

Table 6 shows that for unbalanced data (breast, lung, and liver cancers), the classification metrics (MCCs) of PSO, GA, and COA-HS algorithms were much lower than those of Pre-SDBE (98.07, 97.45, and 96.98 for breast, lung, and liver cancers, respectively). This indicated that the PSO, GA, and COA-HS algorithms did not perform well for unbalanced data.

For the five cancer datasets, whether the data were balanced or unbalanced, Pre-SDBE outperformed the other three algorithms, achieving the highest classification accuracy while identifying fewer number of genes (Table 6). More details are shown in S3 Fig, S5 and S6 Tables.

Table 6. Classification metrics of four optimization algorithms for five cancer datasets.

Algorithm Breast Lung

 #Genes MCC RA F1 SE SP #Genes MCC AUC F1 SE SP

Pre-SDBE 4 98.07 99.42 99.82 99.73 99.12 3 97.45 98.93 99.76 99.71 98.15

PSO a 30 82.98 95.56 98.18 97.00 94.12 29 88.29 98.72 98.70 97.44 100

GA a 18 88.87 98.80 98.78 97.60 100 15 90.88 99.04 99.03 98.08 100

COA-HS a 11 90.93 97.78 99.09 98.50 97.06 8 89.56 98.88 98.87 97.76 100

Liver Colon Prostate

#Genes MCC RA F1 SE SP #Genes AC SE SP #Genes AC SE SP

3 96.98 98.12 99.63 99.75 96.49 2 100 100 100 5 98.99 98.99 98.99

24 62.03 91.87 91.15 83.74 100 11a 96.42a 85.80a 100a 19a 98.04a 91.80a 100a

16 68.30 93.90 93.51 87.80 100 14a 95.16a 84.60a 100a 28a 98.04a 91.80a 100a

9 72.73 95.12 94.87 90.24 100 5a 100a 100a 100a 5a 100a 100a 100a

a Elyasigomari et al. [7]; Pre-SDBE: simplified version of the GSEA–SDBE method; RA: ROC_AUC_score; F1: F1_score; AC: accuracy; SE: sensitivity; SP: specificity; #Genes: number of selected genes.

Note: For unbalanced (breast, lung, and liver) and balanced data (colon and prostate), the performance metrics of the model are different.

3. Response to comment: (Add redundancy assessment measures in addition to MCC.)

Response: I'm very sorry. I can't find other redundancy assessment measures beyond of MCC, but I will work hard in the future.

4. Response to comment: (Provide biological significance of the selected genes by various methods.)

Response: Biological significance of the selected genes as followers.

Relevance of the selected genes to cancer

VEGF-D induces the formation of lymphatics within tumors, thereby facilitating the spread of the tumor to lymph nodes, and promotes tumor angiogenesis and growth [32–36]. TSLP is an interleukin-7 (IL-7)-like cytokine that is involved in the progression of various cancers and is a key mediator of breast cancer progression [37–40]. Human PKMYT1 is an important regulator of the G2/M transition in the cell cycle. Studies have demonstrated that PKMYT1 might be a therapeutic target in hepatocellular carcinoma and neuroblastoma [41–43].

5. Response to comment: (Provide experimental setup for the analysis in more details. What were the criteria used for parameter selection.)

Response: I am very sorry for my negligence. More details were added for the SDBE algorithm, as shown below.

Stage 7: The genes of the list NG were analyzed by backward elimination. At each step of backward elimination, the 10-fold classification mode, e.g., random forest (RF), support vector machine (SVM), k-nearest neighbor (KNN), extreme gradient boosting (XGBoost), and extremely randomized trees (ExtraTrees), and ExtraTrees, was trained and tested to calculate various performance metrics. After each step of backward elimination, the performance metrics were respectively added to the corresponding metric lists. Then, the iteration was terminated and the data were saved. However, if the number of iterations set based on experience was not reached, the metrics lists, which were respectively transposed, and the list NG were sent to stage 3 to start a new iteration.

Stage 8: Mapping analysis of the metrics lists and the list NG was performed and the smallest set of relevant genes needed to achieve the required sample classification performance was determined.

---

## [Editor Report · Decision Letter 2]

14 Jan 2022

GSEA–SDBE: A gene selection method for breast cancer classification based on GSEA and analyzing differences in performance metrics

PONE-D-21-09636R2

Dear Dr. Ai,

We’re pleased to inform you that your manuscript has been judged scientifically suitable for publication and will be formally accepted for publication once it meets all outstanding technical requirements.

Kind regards,

Nguyen Quoc Khanh Le

Academic Editor

PLOS ONE
---

## [Editor Report · Acceptance letter]

18 Apr 2022

PONE-D-21-09636R2 

GSEA–SDBE: A gene selection method for breast cancer classification based on GSEA and analyzing differences in performance metrics 

Dear Dr. Ai:

I'm pleased to inform you that your manuscript has been deemed suitable for publication in PLOS ONE. Congratulations! Your manuscript is now with our production department. 

Kind regards, 

on behalf of

Dr. Nguyen Quoc Khanh Le 

Academic Editor

PLOS ONE